# Inactivated Recombinant Rabies Virus Displaying the Nipah Virus Envelope Glycoproteins Induces Systemic Immune Responses in Mice

**DOI:** 10.3390/vaccines11121758

**Published:** 2023-11-26

**Authors:** Zhengrong Li, Yanting Zhu, Feihu Yan, Hongli Jin, Qi Wang, Yongkun Zhao, Na Feng, Tiecheng Wang, Nan Li, Songtao Yang, Xianzhu Xia, Yanlong Cong

**Affiliations:** 1Laboratory of Infectious Diseases, College of Veterinary Medicine, Key Laboratory of Zoonosis Research, Ministry of Education, Jilin University, Changchun 130122, China; 2Key Laboratory of Jilin Province for Zoonosis Prevention and Control, Changchun Veterinary Research Institute, Chinese Academy of Agricultural Sciences, Changchun 130122, China

**Keywords:** Nipah virus, envelope glycoproteins, recombinant rabies virus, immune response, vaccine

## Abstract

Nipah virus (NiV) causes severe, lethal encephalitis in humans and pigs. However, there is no licensed vaccine available to prevent NiV infection. In this study, we used the reverse genetic system based on the attenuated rabies virus strain SRV9 to construct two recombinant viruses, rSRV9-NiV-F and rSRV9-NiV-G, which displayed the NiV envelope glycoproteins F and G, respectively. Following three immunizations in BALB/c mice, the inactivated rSRV9-NiV-F and rSRV9-NiV-G alone or in combination, mixed with the adjuvants ISA 201 VG and poly (I:C), were able to induce the antigen-specific cellular and Th1-biased humoral immune responses. The specific antibodies against rSRV9-NiV-F and rSRV9-NiV-G had reactivity with two constructed bacterial-like particles displaying the F and G antigens of NiV. These data demonstrate that rSRV9-NiV-F or rSRV9-NiV-G has the potential to be developed into a promising vaccine candidate against NiV infection.

## 1. Introduction

Nipah virus encephalitis is a zoonotic infectious disease caused by Nipah virus (NiV) [1]. Since its initial emergence in Malaysia in 1998, NiV outbreaks have spread to subtropical areas, including Malaysia, Bangladesh, the Philippines, Singapore, and India [2]. NiV spreads via the main routes of fruit bat-pig-human and human-to-human transmission, resulting in respiratory symptoms and fatal encephalitis in humans and pigs [3]. Recently, NiV encephalitis has been included in the WHO blueprint list of priority diseases for research due to its high lethality and destructiveness [4]. Vaccines are the most cost-effective and equitable way to combat and eradicate infectious diseases. However, there is currently no vaccine available for NiV encephalitis.

Recombinant viral vectors have been in use for over 40 years for the delivery of specific pathogen antigens [5]. For NiV, a few viral vectors, including canary pox virus, smallpox virus, adeno-associated virus, chimpanzee adenovirus, Newcastle disease virus, measles virus, and rabies virus (RABV) have been used in preclinical vaccine studies [6]. A recombinant vesicular stomatitis virus vaccine candidate against NiV has entered Phase I clinical trials [7], suggesting the potential of viral vector vaccines for the prevention of NiV encephalitis. Although substantial progress has been made toward the development of NiV encephalitis vaccines, novel vaccine development efforts are essential because of the complexity of NiV pathogenesis and immunity.

Compared to other virus vectors, RABV has many advantages as a vaccine vector, such as a high yield (up to 10^8^ TCID_50_/mL) of reverse genetically rescued RABV, ease of large-scale culture, low level of pre-existing antibodies in vivo, and high level of expression of foreign proteins, which is very promising. Currently, RABV is being used with great promise in the development of novel vaccines against a variety of emerging virus, such as Ebola virus [8], Lassa fever virus [9], Middle East respiratory syndrome virus [10], and Zika virus [11].

In the NiV phylogenetic tree, two distinct lineages exist, including the Malaysian lineage and Bangladesh lineage. Compared to the Malaysian lineage, the Bangladesh lineage, which is endemic in Bangladesh and India, is associated with more severe respiratory disease and high mortality [6]. Among the six structural proteins encoded by NiV, the fusion (F) and glycoprotein (G) proteins attached to the surface of the envelope are the two most important immunoprotective antigens capable of inducing neutralizing antibodies [12,13,14]. In the present study, the F and G proteins of NiV from Bangladesh were selected as target proteins, and the SRV9 strain of RABV was used as a vector to construct two recombinant NiV vaccines to compare the immunodominance of F and G proteins and their synergistic immunization effects.

## 2. Materials and Methods

### 2.1. Identification of Recombinant RABVs Displaying the F or G proteins of NiV

#### 2.1.1. Rescue of Recombinant RABVs Displaying the F or G Proteins of NiV

Based on the established reverse genetic system of attenuated RABV strain SRV9 [15], the synthesized open-reading frames of the F and G genes of NiV (GenBank No.: AY988601.1) were inserted between the P and M genes of pSRV9, respectively. As shown in Figure 1, the two constructed plasmids expressing the NiV genes were designated as pSRV9-NiV-F and pSRV9-NiV-G, respectively. BSR cells (a clone of baby hamster kidney-21 cells) in a 6-well plate were transfected with pSRV9-NiV-F or pSRV9-NiV-G together with the RABV helper plasmids (pcDNA3.1-N, pcDNA3.1-P, pcDNA3.1-L, and pcDNA3.1-G) using TransIT^®^-LT1 (Mirus, Madison, WI, USA) to obtain the recombinant RABVs. The mixture of cells and supernatant were collected on day 7 post-transfection.

#### 2.1.2. Direct Immunofluorescence

To determine whether the recombinant RABVs were successfully rescued, a direct immunofluorescence assay (dIFA) was conducted as described previously [15,16]. Briefly, the recombinant RABV (50 μL/well) was added to BSR cells cultured in a 96-well cell culture plate. After 48 h, the cells were fixed with 80% cold acetone for 1 h at −20 °C and then washed three times with phosphate buffered saline with 0.05% Tween-20 (PBST). The fluorescein isothiocyanate (FITC)-labeled anti-RABV N-protein monoclonal antibody (mAb) (Fujirebio, Melvin, PA, USA) at 1:200 dilution was added to the plate and incubated at 37 °C for 1 h. The fluorescence focus was observed under a fluorescence microscope (Leica, Weztlar, Germany).

#### 2.1.3. Indirect Immunofluorescence

To confirm whether the recombinant RABV can display the F or G proteins of NiV, an indirect immunofluorescence assay (iIFA) was performed as described previously [17,18]. Briefly, BSR cells cultured in 96-well cell culture plates infected with each of the recombinant RABVs (50 μL/well) were fixed with 80% cold acetone for 1 h at −20 °C and then incubated for 1 h at room temperature (RT) for NiV staining using a 1:500 dilution of NiV F or NiV G mAb (MyBiosource, San Diego, CA, USA). Detection was performed with a Cy3-conjugated goat anti-mouse IgG (Beyotime, Shanghai, China) as a secondary antibody at 1:1000 dilution for 1 h at RT. Fluorescence was observed with a fluorescence microscope (Leica, Weztlar, Germany).

#### 2.1.4. Sucrose Density Gradient Centrifugation

The recombinant RABVs were purified by the sucrose density gradient centrifugation [19]. Briefly, the recombinant virus inactivated with beta-propiolactone (BPL) (Serva Electrophoresis GmbH, Heidelberg, Germany) was centrifuged at 3000 rpm for 15 min to remove the cell fragments. The harvested supernatant was supplemented with 1 M zinc acetate (Sangon Biotech, Shanghai, China) at a volume ratio of 50:1. After standing for 1 h at 4 °C, the sample was centrifuged at 10,000 rpm for 30 min at 4 °C. The precipitate was dissolved in saturated EDTA solution (Sangon Biotech, Shanghai, China) overnight at 4 °C and then centrifuged at 21,000 rpm for 90 min through 20%, 30%, 40%, and 55% sucrose density gradients. The recombinant RABV between 40 and 55% sucrose gradient was harvested, dissolved in sodium Chloride-Tris-EDTA (STE) buffer (0.15 M NaCl, 0.001 M EDTA, and 0.01 M Tris-base, pH 7.4), and centrifuged at 25,000 rpm for 90 min to remove the sucrose. The recombinant RABV precipitate was resuspended and dissolved in PBS and stored at −80 °C.

#### 2.1.5. Western Blotting

The purified recombinant RABV was further identified to confirm the expression of the F or G proteins of NiV by SDS-PAGE and Western blot as described previously [20]. The recombinant RABV was added to SDS-PAGE loading buffer (CWBIO, Taizhou, China), and viruses denatured in boiling water for 5 min were loaded onto a 10% polyacrylamide gel. For total protein analysis, the polyacrylamide gel was stained with Coomassie Brilliant Blue for 20 min, and the gel was then immersed in Coomassie Brilliant Blue destaining solution to destain until the gel background turned white. For Western blotting, the target protein was transferred to a 0.45 µm pore-size nitrocellulose (NC) membrane by the semi-dry transmembrane method. After blocking in 5% skimmed milk at RT for 2 h, the membrane was incubated overnight with the mAb of NiV glycoproteins (Mybiosource, Southern California, San Diego, CA, USA) at 1:1000 dilution and then incubated at RT for 1 h with the horseradish peroxidase (HRP)-conjugated goat anti-mouse IgG (CWBIO, Taizhou, China) at 1:20,000 dilution. Enhanced chemiluminescence (ECL) color developer was dropped onto the NC membrane for exposure and color development.

#### 2.1.6. Transmission Electron Microscopy

The morphology of recombinant RABV was observed by transmission electron microscope (TEM). Briefly, 20 μL of recombinant RABV was dropped onto the glowing copper grid and incubated at RT for 2 min. After allowing the copper grid to air dry, the front of the copper grid was stained with 2% phosphotungstic acid at RT for 1 min. The recombinant RABV particles were observed by TEM [21].

#### 2.1.7. Immunoelectron Microscopy

The presence of F or G proteins of NiV on the surface of RABV was observed by immunoelectron microscopy (IEM) as previously described [22]. Briefly, the adsorption step of recombinant RABV on a copper grid was the same as in TEM. The copper grid was placed in 4% paraformaldehyde to fix for 3–5 min, and then washed with PBS and blocked with 5% skim milk for 1 h. The samples were double-stained with mouse anti-RABV-G mAb (1:30) (Sigma, St. Louis, MO, USA) and rabbit anti-NiV-F_etc_ or NiV-G_etc_ polyclonal antibody (pAb) (homemade) (1:30) for 1 h, followed by incubation with donkey anti-mouse IgG antibody (18 nm gold, 1:20) (Abcam, Cambridge, MA, USA) and donkey anti-rabbit IgG antibody (6 nm gold, 1:40) (Abcam, Cambridge, MA, USA) for 1 h. The copper grid was stained with 2% phosphotungstic acid for 1 min, and the recombinant RABV particles were examined under TEM.

#### 2.1.8. Multi-Step Growth Kinetic Curve

BSR cells cultured in a 6-well cell culture plate were infected with the recombinant RABV at an MOI of 0.5, and 300 μL of supernatants were collected every 24 h for 4 days. The viral TCID_50_ was determined to plot the growth kinetic curve of the recombinant RABV using the Reed–Muench method as described [23].

### 2.2. Analysis of Recombinant RABVs Displaying the F or G Proteins of NiV

#### 2.2.1. Preparation of Immunogen

The immunogen supplemented with adjuvants was prepared at a volume ratio of 350 μL of diluted purified recombinant RABV, 550 μL of ISA 201 VG (Seppic, Paris, France), and 100 μL of poly (I:C) (0.2 mg/mL) (Sigma, St. Louis, MO, USA) per 1 mL and then preheated at 31 °C for 10 min. After emulsion, the immunogen was allowed to stand at 20 °C for 1 h. A drop of immunogen was added to the cell counting plate and observed under a light microscope. If there were no large oil droplets, the preparation was considered complete.

#### 2.2.2. Immunization of Mice

To evaluate the immunogenicity of recombinant RABV, SPF female BALB/c mice aged 4–6 weeks old were purchased from Beijing Vital River Laboratory Animal Technology Co., Ltd. (Beijing, China). Before the experiment began, they were raised for 2 weeks to adapt to the environment. They were then randomly divided into four groups of 5 mice each. Mice in the three experiment groups were immunized intramuscularly with 20 μg/100 μL of immunogen, while the control group was immunized with 100 μL of PBS. After the primary immunization, booster immunizations were performed twice at weeks 3 and 6, respectively. Sera were collected at weeks 2, 5, and 8 after the primary immunization and heat-inactivated at 56 °C for 30 min.

#### 2.2.3. Splenic Lymphocyte Proliferation Assay

To investigate the immunological memory effect of splenic lymphocytes from mice immunized with the recombinant RABV, we performed a splenic lymphocyte proliferation assay as described previously [24]. Briefly, 5 × 10^4^ cells//well of lymphocyte suspensions from mice on 14 days after the third immunization were added to a 96-well cell culture plate and then incubated at 37 °C with a final concentrate of 0.25 μg/mL of F and G alone or in combination. After 44 h, 10 μL/well of CCK-8 (Solarbio, Beijing, China) was added, and the optical density (OD) value at 450 nm was measured at 4 h after co-incubation. The proliferation curves of splenic lymphocytes were plotted according to the stimulation index (SI) formula: SI = (OD_sample well_ − OD_blank well_)/(OD_negative well_ − OD_blank well_) ratio.

#### 2.2.4. Cytokine Detections by ELISA and ELISpot

Commercial mouse cytokine ELISA kits (DAKEWE, Shenzhen, China) were used to detect cytokines in the supernatants of splenocytes stimulated with the recombinant RABV. Briefly, 5 × 10^6^ cells//well of splenocyte suspension and a final concentrate of 0.25 μg/mL of F and G alone or in combination were added to 6-well cell culture plates. After incubation for 72 h, IFN-γ, TNF-α, IL-2, IL-4, IL-6, and IL-10 in the supernatants were quantified. Meanwhile, IFN-γ and IL-4 secreted by splenic lymphocytes after stimulation with the recombinant RABV were detected using the mouse cytokine ELISpot kit (Mabtech, Nacka, Sweden) according to the instructions. Briefly, 5 × 10^5^ cells//well of splenocyte suspension and 10 μg/mL of recombinant RABV were simultaneously added to a 96-well ELISpot plate. After incubation for 24 h, primary antibodies against IFN-γ or IL-4 (1:1000) were added and incubated for 2 h. After three washes with PBST, the plate was incubated with streptavidin-HRP at 1:1000 dilution for 1 h. After tetramethylbenzidine (TMB) color development, multiple analyses were performed at the single cell level using RAWspot technology on a Mabtech IRIS FluoroSpot/ELISpot reader.

#### 2.2.5. Indirect ELISA

To determine the titer and subtype of specific antibodies against NiV in serum, we developed an indirect enzyme-linked immunosorbent assay (iELISA) [25]. In brief, 0.5 μg/mL of F or G proteins of NiV were coated onto ELISA plates overnight at 4 °C. After three washes with PBST, the plate was blocked with 3% bovine serum albumin (BSA) for 2 h. After three washes with PBST, 50 μL/well of the diluted inactivated serum samples were added and incubated for 1 h at 37 °C. After washing, a 1:10,000 dilution of HRP-conjugated goat anti-mouse IgG (CWBIO, Taizhou China), a 1:5000 dilution of HRP-conjugated goat anti-mouse IgG1 (Southern Biotech, Birmingham, AL, USA), or a 1:5000 dilution of HRP-conjugated goat anti-mouse IgG2a (Southern Biotech, Birmingham, AL, USA) was added and incubated at 37 °C for 1 h. After TMB color development, the OD_450nm_ value of each well was read. A test sample/negative sample with an OD_450nm_ greater than 2 was considered positive.

### 2.3. Preparation of the Bacterial-Like Particles Displaying the F or G Proteins of NiV

#### 2.3.1. Rescue of Recombinant Baculovirus

To identify the reactivity of immune serum in a more intuitive manner, we constructed two bacterial-like particles (BLPs) that displayed the F or G proteins of NiV. In brief, the synthesized extracellular domain of the F or G genes of NiV (GenBank No. AY988601.1) fused to the anchor protein PA (GenBank No. U17696) of *Lactococcus lactis* (*L. lactis*) was codon-optimized based on the codon preference of insect cells and cloned into the pFastBac1 transfer vector (Thermo Fisher Scientific, Waltham, MA, USA). These two transfer vectors, pFB1-F_etc_ and pFB1-G_etc_, were transformed into DH10Bac competent cells containing the baculovirus genome for blue and white screening. Sf9 cells were transfected with recombinant bacmids using Cellfectin^TM^ II (Invitrogen, Waltham, MA, USA) to obtain the recombinant baculoviruses rBV-NiV-F_etc_ and rBV-NiV-G_etc_. The rescued recombinant baculovirus was inoculated at a volume ratio of 3% into the monolayer of Sf9 cells cultured on a 60 mm diameter cell culture dish. After 96 h, Sf9 cells were harvested and subjected to ultrasonic fragmentation to obtain the fusion proteins F_etc_-PA or G_etc_-PA.

#### 2.3.2. Preparation of Gram-Positive Enhancer Matrix Particles

*L. lactis* MG1363 cultured in M17 broth medium was harvested and resuspended in a 10% trichloroacetic acid solution (Sigma, St. Louis, MO, USA). After boiling for 30 min and washing five times with 0.01 M PBS, the resulting Gram-positive enhancer matrix (GEM) particles were diluted to 1 unit (U), where 1 U was defined as 2.5 × 10^9^ GEM particles.

#### 2.3.3. Construction of the Bacterial-Like Particles

A total of 8 mL of the fusion proteins F_etc_-PA or G_etc_-PA expressed by rBV-NiV-F_etc_ or rBV-NiV-G_etc_ was bound with 1 U GEM on a shaker at RT for 1 h. After centrifugation at 8000 rpm for 10 min at 4 °C, the precipitate was washed five times with an equal volume of 0.01 M PBS to obtain the fusion protein-GEM complex, which is BLP.

### 2.4. Cross-Reaction between the Recombinant RABVs and the BLPs

To determine the reactivity between the specific antibodies against NiV in the serum after the third immunization and the NiV glycoproteins displayed on the BLP surface, IEM observation was performed as described above. Briefly, 20 μL of acidified GEM particles, BLP-NiV-F_etc_, or BLP-NiV-G_etc_ were dropped onto the parafilm. The copper grid with parafilm was placed face-down and incubated at RT for 15 min. After washing with PBS, the sample was fixed in 4% paraformaldehyde (PFA) for 3–5 min and then blocked with 5% skim milk for 1 h. The copper grid was incubated with the serum of mice immunized with recombinant RABVs (1:30) as the primary antibody and donkey anti-mouse IgG antibody (18 nm gold, 1:20) as the secondary antibody (Abcam, Cambridge, MA, USA) at RT for 1 h each. The sample was stained with 2% phosphotungstic acid at RT for 1 min, and the colloidal gold labeling was observed under TEM.

### 2.5. Statistical Analysis

All statistical analyses were performed using the GraphPad Prism 8.0.1 statistical software. Experimental data were analyzed by the One-way or Two-way ANOVA methods. The mean and standard deviation (mean ± SD) were used to express the experimental data. Statistical differences were indicated by an asterisk. * *p* < 0.05; ** *p* < 0.01; *** *p* < 0.001.

## 3. Results

### 3.1. Identification of Recombinant RABVs Displaying the F or G Proteins of NiV

Using the RABV reverse genetic system, the recombinant plasmids pSRV9-NiV-F and pSRV9-NiV-G were identified by PCR and digested with *Bsiw I* and *Pme I* restriction endonuclease enzymes (the primers of F and G genes of NiV are shown in Table 1). The specific F (1641 bp) or G (1809 bp) genes were inserted into the RABV expression vector pSRV9 as expected. BSR cells were then transfected with pSRV9-NiV-F or pSRV9-NiV-G together with the RABV helper plasmids to obtain the recombinant viruses rSRV9-NiV-F or rSRV9-NiV-G. To characterize whether the recombinant RABVs were successfully rescued, we first identified the specific protein of RABV by dIFA with FITC-labeled anti-RABV N-protein mAb. The results showed that green fluorescence appeared in the cytoplasm after BSR cells were infected with rSRV9-NiV-F or rSRV9-NiV-G (Figure 2A,B). The subsequent iIFA showed specific red fluorescence when using mouse anti-NiV-F or NiV-G mAb as the primary antibody and Cy3-labeled goat anti-mouse IgG as the secondary antibody (Figure 3). To test whether the introduction of foreign genes would alter the growth characteristics of RABV, a multi-step growth kinetic curve was plotted at an MOI of 0.5. As shown in Figure 4A, there was no significant difference between the recombinant RABVs and the RABV vector virus (*p* > 0.05). Also, the SDS-PAGE results clearly showed the five RABV structural proteins expressed by the RABV vector virus and the F protein of NiV (~60 kDa) (Figure 4B,C). Since the G proteins of RABV and NiV were similar in molecular weight (~67 kDa), the two G protein bands appeared to overlap, which was confirmed by Western blots (Figure 4D). Under TEM, the bullet-shaped, enveloped virus particles were clearly visible (Figure 4E–G). To determine the distribution of the NiV glycoproteins on the recombinant RABVs, we performed an IEM. As shown in Figure 4H–J, gold particles specifically recognizing the F or G proteins of NiV were present on the surface of the recombinant RABVs.

### 3.2. The Immunogenicity of Recombinant RABVs Displaying the F or G Proteins of NiV

To evaluate the effect of recombinant RABVs on the bodyweight of mice, all mice were weighed every other day after immunization, and the relative bodyweight of the mice was calculated. The results showed there was no statistical difference between the recombinant RABV-immunized groups and the control group (*p* > 0.05). One week after the third immunization, the immunological memory and response effects of splenic lymphocytes were evaluated. The SI values of the mice in the immunized groups were all significantly higher than those in the control group (*p* < 0.001), and the SI values of the co-immunized group were significantly higher than those in the rSRV9-NiV-G- and rSRV9-NiV-F-immunized group (*p* < 0.01–0.001) (Figure 5A). After stimulation of splenic lymphocytes with rSRV9-NiV-F and rSRV9-NiV-G alone or in combination, the numbers of IFN-γ and IL-4-responsive spot-forming cells (SFCs) in the immunized mice were significantly higher than those in the control group (*p* < 0.01–0.001) (Figure 5B,C). To further evaluate the immune effect of recombinant RABVs, we measured the levels of cytokines secreted by splenic lymphocytes. The results showed that the Th1-type cytokine (TNF-α) and Th2-type cytokines (IL-4, IL-6, IL-10) secreted by splenic lymphocytes were increased (*p* < 0.01–0.0001) (Figure 5D–I). These data demonstrate that rSRV9-NiV-F and rSRV9-NiV-G can induce the proliferation of splenic lymphocytes and the antigen-specific cellular immune response in mice. Meanwhile, the specific serum IgG titers were determined after each immunization. The results showed that the IgG levels in the rSRV9-NiV-G-immunized group were higher than those in the other groups at 14 days post-immunization (*p* < 0.05–0.01) and higher than those in the rSRV9-NiV-F group and the combined group at 56 days post-immunization (*p* < 0.01–0.001). At 35 days after immunization, the IgG levels were higher in all groups than in the control group (*p* < 0.01–0.001) (Figure 5J). In addition, the IgG subtype showed that the ratio of IgG2a/IgG1 produced after the last immunization was greater than 1 (Figure 5K).

### 3.3. The Reactivity of the Immune Sera against Recombinant RABVs Displaying the F or G Proteins of NiV

To verify the reactivity of the specific antibodies against NiV in the serum after immunization, BLP-NiV-F_etc_ and BLP-NiV-G_etc_, which display the extracellular domain of the F or G proteins of NiV, were used as antigens to co-incubate with the serum of mice immunized with rSRV9-NiV-F and rSRV9-NiV-G. The reactivity of the serum antibodies was characterized by IEM. As shown in Figure 6, the gold particles were adhered to the surface of BLPs.

## 4. Discussion

The context of the COVID-19 global pandemic has increased our focus on research into other potential pandemic pathogens. The susceptibility to humans, ability to spread from person to person, and ease of mutation make NiV a potential global pandemic risk [1]. One of the most important strategies for preventing the infection and transmission of NiV is the use of vaccines. Extensive NiV vaccine research, including preclinical studies in numerous animal models, has shown that recombinant viral vector vaccines and subunit vaccines provide protective immunity [26]. However, to date, there is no NiV vaccine available.

In this study, we used the RABV vector to recombine two immunoprotective proteins, the F and G of NiV, to evaluate their immunogenicity. In the process of rescuing rSRV9-NiV-F and rSRV9-NiV-G, it was found that although the G protein of NiV is a type II glycoprotein with a tetrameric structure, which is different from the G protein (type I glycoprotein) of RABV, it does not affect the assembly of rSRV9-NiV-G. Although the molecular sizes of the G proteins of NiV and RABV are similar, which cannot be distinguished by SDS-PAGE (Figure 4B), the identification by iIFA and Western blot confirmed that the G protein of NiV was expressed by the SRV9 vector (Figure 3 and Figure 4D). Furthermore, the insertion of F or G proteins of NiV not only had no effect on the morphology of the RABV particles (Figure 4F,G), but the 6 nm-diameter gold-labeled particles were also evenly distributed on the surface of the rhabdovirus (Figure 4H,I), indicating that two glycoproteins of NiV were displayed on the surface of RABVs, respectively. Most importantly, the insertion of F or G proteins of NiV had no effect on the growth characteristics of the vector virus SRV9 (Figure 4A). Taken together, these results demonstrate that rSRV9-NiV-F and rSRV9-NiV-G have the potential to be vaccine candidate seeds.

Cellular and humoral immune responses are important indicators for evaluating vaccine effectiveness. With regard to the cellular immune response to rSRV9-NiV-F and rSRV9-NiV-G, the results of the splenic lymphocyte proliferation assays, ELISpot, and ELISA showed that both of them induced the proliferation and activation of splenic lymphocytes (Figure 5D–K), indicating the immunological memory effect of splenic lymphocytes obtained from the mice immunized with rSRV9-NiV-F and rSRV9-NiV-G alone or in combination. Although the cytokine levels in the co-immunized group were much higher than those in the rSRV9 NiV-F- or rSRV9 NiV-G-immunized group, the IgG titer in the co-immunized group was lower than that in the single immunized group after three immunizations (Figure 5J). The antibody subtype analysis indicated that the IgG induced by rSRV9-NiV-F or rSRV9-NiV-G was predominantly of the IgG2a subtype (Figure 5K), which favors the Th1-type cell immune response. It has been shown that this response is usually considered the most potent in activating effector responses and plays a dominant role in antiviral immunity [27]. In addition, Th1-type antibodies are known to contribute to non-neutralizing effects, such as NK cell-mediated antibody-dependent cytotoxicity or antibody-dependent cellular phagocytosis mediated by macrophages and monocytes [28,29,30,31].

Regardless, the viral challenge is the most direct and effective way to evaluate vaccine candidates. In this process, animal models are essential for testing and validating the potential of a vaccine. Several established animal models of NiV infection and pathogenesis are now available, including pigs, cats, mice, ferrets, guinea pigs, squirrel monkeys, hamsters, ferrets, African green monkeys, and horses [32,33]. Some of the representative experiments, including canarypox and cowpox virus vector vaccines encoding NiV glycoproteins, showed protection against NiV in hamster and pig models [34,35]; a recombinant VSV vector vaccine expressing NiV-G protein protected ferrets against NiV challenge [36]; another replication-defective VSV vector-based NiV vaccine protected Syrian hamsters against lethal challenge [37]; and a recombinant measles virus vector vaccine expressing the G protein of NiV was effective in the African green monkey model [38]. Unfortunately, we were unable to conduct animal immunoprotection tests in this study due to the lack of a NiV strain and a biosafety level 4 (BSL-4) laboratory. This is one of the shortcomings of this study. To overcome these problems, we constructed two BLPs displaying the F or G antigens of NiV to evaluate the reactivity of immune sera against rSRV9-NiV-F or rSRV9-NiV-G. The IEM results indicated that the specific sera against rSRV9-NiV-F and rSRV9-NiV-G had reactivity (Figure 6).

In conclusion, the inactivated recombinant RABVs displaying the F or G proteins of NiV were able to induce the antigen-specific cellular and Th1-biased humoral immune responses. Moreover, the immune sera against rSRV9-NiV-F and rSRV9-NiV-G had reactivity. Our study provides promising ideas and useful data references for the future development of NiV encephalitis vaccines.

## Figures and Tables

**Figure 1 vaccines-11-01758-f001:**
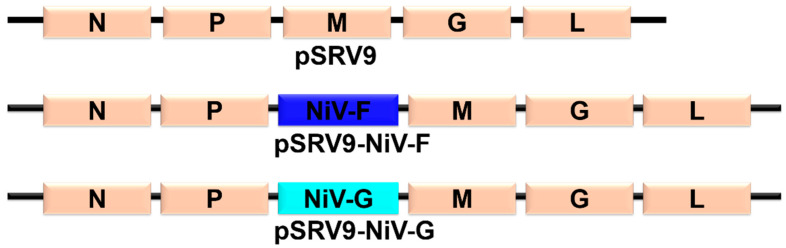
Schematic diagram of the construction of a recombinant RABV vector expressing the NiV glycoproteins F or G.

**Figure 2 vaccines-11-01758-f002:**
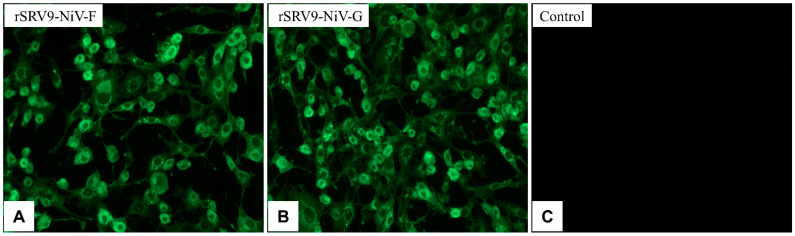
Identification of recombinant RABVs expressing the F or G proteins of NiV by direct immunofluorescence using an FITC-labeled anti-RABV N-protein mAb. Panel (**A**): rSRV9-NiV-F; panel (**B**): rSRV9-NiV-G; panel (**C**): negative control for virus uninfected cells. Magnification: 200×.

**Figure 3 vaccines-11-01758-f003:**
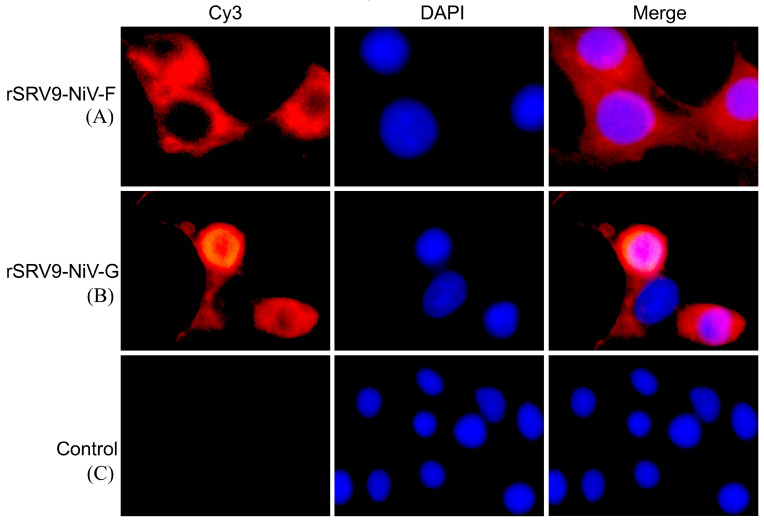
Identification of recombinant RABVs expressing the F or G proteins of NiV by indirect immunofluorescence using the mouse anti-NiV-F or NiV-G mAb as the primary antibody and the Cy3-labeled goat anti-mouse IgG as the secondary antibody. Panel (**A**): rSRV9-NiV-F; panel (**B**): rSRV9-NiV-G; panel (**C**): control. Magnification: 400×.

**Figure 4 vaccines-11-01758-f004:**
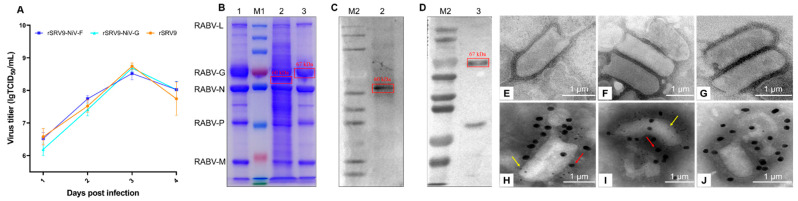
Proliferation and identification of recombinant RABVs. Panel (**A**): multi-step growth kinetic curves of recombinant RABV. BSR cells were infected with rSRV9-NiV-F, rSRV9-NiV-G, or rSRV9 at an MOI of 0.5. The logarithmic function lg is a logarithm (common logarithm) with base 10, e.g., lg 10 = 1. Lg is log10.The significance of difference was analyzed by a Two-Way ANOVA test in GraphPad Prism 8.0.1 software, followed by a Least Significance Difference test. Panel (**B**): identification of recombinant RABVs expressing the F or G proteins of NiV by SDS-PAGE. Panel (**C**,**D**): identification of recombinant RABVs expressing the F or G proteins of NiV by Western blot. Lane M1: molecular size markers from top to bottom: 250, 130, 100, 70, 55, 35, 25, 15, and 10 kDa, respectively; lanes M2: molecular size markers from top to bottom: 180, 130, 100, 70, 55, 40, 35, 25, 15, and 10 kDa, respectively; lane 1: rSRV9; lane 2: rSRV9-NiV-F; lane 3: rSRV9-NiV-G. Panels (**E**–**G**): rSRV9-NiV-F, rSRV9-NiV-G, and rSRV9 under transmission electron microscope (TEM). Panels (**H**–**J**): rSRV9-NiV-F, rSRV9-NiV-G, and rSRV9 under immunoelectron microscope (IEM). Red arrows: 18 nm gold-labeled antibody. Yellow arrows: 6 nm gold-labeled antibody. Bar: 1 μm.

**Figure 5 vaccines-11-01758-f005:**
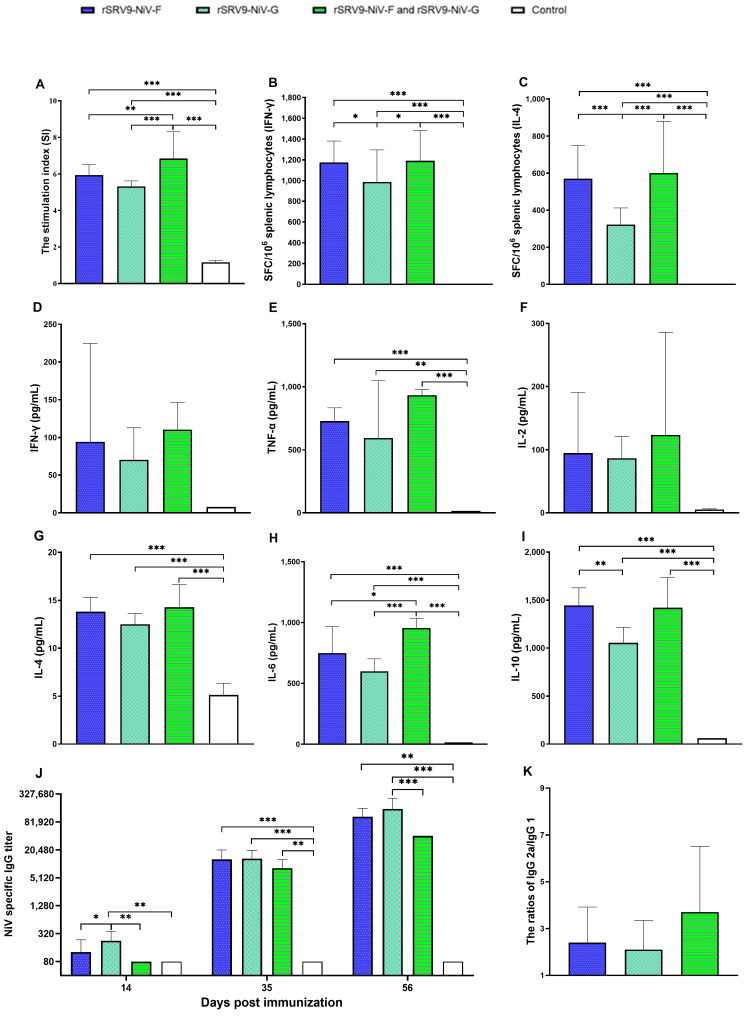
Effects of rSRV9-NiV-F and rSRV9-NiV-G alone or in combination on splenic lymphocytes. Panel (**A**): the stimulation index was detected using a CCK-8 assay. Panels (**B**,**C**): the spot numbers of IFN-γ and IL-4. Panels (**D**–**I**): ELISA results of IFN-γ, TNF-α, IL-2, IL-4, IL-6, and IL-10 secreted by splenic lymphocytes after stimulation. Panel (**J**): specific IgG antibody titers determined by ELISA. Panel (**K**): identification of IgG subclasses. The significance of difference was analyzed by a One-Way ANOVA test in GraphPad Prism 8.0.1 software, followed by a Least Significance Difference test. * *p* < 0.05; ** *p* < 0.01; *** *p* < 0.001.

**Figure 6 vaccines-11-01758-f006:**
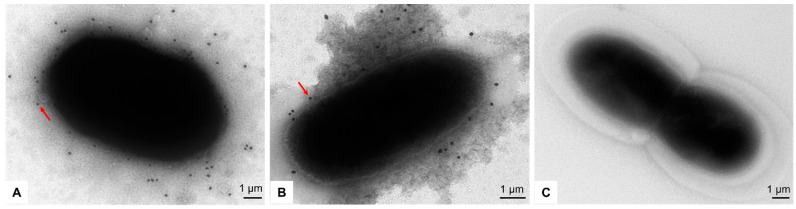
IEM characterization of the reactivity of recombinant RABV immune sera with BLPs displaying the F or G proteins of NiV. BLPs probed first with mouse antiserum raised against rSRV9-NiV-F or rSRV9-NiV-G and then with gold-labeled donkey anti-mouse IgG antibody. Panel (**A**): BLP-NiV-F_etc_ recognized by mouse antiserum raised against rSRV9-NiV-F; Panel (**B**): BLP-NiV-G_etc_ recognized by mouse antiserum raised against rSRV9-NiV-G; Panel (**C**): the reactivity of BLP with serum from non-immune group. Red arrows: 18 nm gold-labeled donkey anti-mouse IgG antibody. Bars: 1 μm.

**Table 1 vaccines-11-01758-t001:** The PCR primers used to amplify the F and G genes of NiV. The sequences of restriction enzyme sites are underlined and italicized.

Primers	Sequence (5′ → 3′)
NiV/F-F	GAC*CGTACG*GCCACCATGGCAGTTATACTTAACAAGAGAT
NiV/F-R	GGC*GTTTAAAC*CTACGTCCCAATGTAATAGAGATCC
NiV/G-F	TAG*CGTACG*GCCACCATGCCGACAGAAAGCAAGAAAGTTA
NiV/G-R	CAA*GTTTAAAC*TTATGTACATTGCTCTGGTATCTTA

## Data Availability

Data is contained within the article.

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
