# Peer review of "Inactivated Recombinant Rabies Virus Displaying the Nipah Virus Envelope Glycoproteins Induces Systemic Immune Responses in Mice"

_vaccines, 2023, doi:10.3390/vaccines11121758_

Round 1

Reviewer 1 Report

Comments and Suggestions for Authors

This manuscript briefly reported the successful construction of the two recombinant rSRV9-NiV-F and rSRV9-NiV-G based on the attenuated rabies virus strain SRV9, and vaccination of BALB/c mice with the recombinant viruses. The inactivated viruses were confirmed to induce the antigen-specific cellular and Th1-biased humoral immune response against rabies virus and Nipah virus. These proved that rSRV9-NiV-F or rSRV9-NiV-G are promising candidate vaccine for the control of NiV infection. This report is interesting and attractive for the prevention of rabies and Nipah virus encephalitis, and it is acceptable for publication after the following problems are solved.

1.      The manuscript exists some grammatical mistakes, and it   should be carefully revised with the help of native expert.     

2.      The authors claimed that the recombinant immunization induced the antigen-specific cellular immune response, but they did not provide the results about the specific CD4+ and CD8+ T cell immune reaction to the specific NiV and RABV antigen, which should be added in their revised manuscript.

3.      The key words should be revised to reflect the full content of manuscript.  

Comments on the Quality of English Language

There are some grammatical mistakes in the manuscript , and it  should be carefully revised with the help of native expert.

Author Response

  1. The manuscript exists some grammatical mistakes, and it should be carefully revised with the help of native expert.

A: The grammar of this manuscript has been revised by native English-speaking professionals.

  1. The authors claimed that the recombinant immunization induced the antigen-specific cellular immune response, but they did not provide the results about the specific CD4+ and CD8+ T cell immune reaction to the specific NiV and RABV antigen, which should be added in their revised manuscript.

A: Due to research funding constraints, we have not been able to utilize techniques such as flow cytometry to detect the immune response of CD4+ and CD8+ T lymphocytes. However, ELISpot is an immunoenzyme technique that quantitatively detects individual cells. Under the specific antigen stimulation, the cytokines secreted by cells will be captured by antibodies. Thus, it can represent the immune response of lymphocytes.

  1. The key words should be revised to reflect the full content of manuscript.

A: We have revised the keywords.

Reviewer 2 Report

Comments and Suggestions for Authors

In this manuscript, Li and colleagues constructed two recombinant rabies viruses that expressed the NiV envelope glycoproteins F and G, respectively. The F or G protein was presented on the envelope of the virus. They evaluated the cellular immune and humoral immune responses of inactivated recombinant viruses by immunizing mice. This work provided insights into developing vaccines against NiV infection. However, after reviewing the manuscript, I have several concerns about the current version of the manuscript. Before being accepted for publication, these concerns should be revised or discussed.

Major:

1. The most direct method to evaluate the immunological protection effect of vaccine is animal challenge test. It is acceptable that animal challenge test is not carried out because of dangerous pathogens. However, the absence of virus-neutralizing antibody testing in this study is an important shortcoming. If the authentic NiV cannot be used for neutralizing antibody detection due to biosafety considerations, the pseudovirus system can also be used for neutralizing antibody detection.

2. The detection items of cellular immunity in this study are not specific to NiV F or G protein. If there are F and G protein specific cellular immunity indicators, it can better reflect the effect of recombinant virus vaccine for NiV.

3. English writing needs to be improved.

Minor

1. It is suggested to re-label Figure 1, Figure 2, and Figure 4 to improve the readability of the figure. so that the reader does not have to look for the notes repeatedly to understand the figures.

2. In Fig.3, “Anti-NiV F/G protein” should be marked as “Cy3”.

3. Line 65, Line 241: IFA is usually an abbreviation of immunofluorescence assay. In this manuscript, authors adopted indirect immunofluorescence assay and direct immunofluorescence assay respectively. I suggest that iIFA and dIFA should be adopted here.

4. Line 175: “ELSIA” should be “ELISA”.

Author Response

Major:

  1. The most direct method to evaluate the immunological protection effect of vaccine is animal challenge test. It is acceptable that animal challenge test is not carried out because of dangerous pathogens. However, the absence of virus-neutralizing antibody testing in this study is an important shortcoming. If the authentic NiV cannot be used for neutralizing antibody detection due to biosafety considerations, the pseudovirus system can also be used for neutralizing antibody detection.

A: Due to biosafety considerations, we have not performed a corresponding neutralization assay, and the experiments in this paper allowed us to systematically assess the immunological effects of the vaccine.

  1. The detection items of cellular immunity in this study are not specific to NiV F or G protein. If there are F and G protein specific cellular immunity indicators, it can better reflect the effect of recombinant virus vaccine for NiV.

A: We have tried to construct pseudovirus systems for NiV, but were unsuccessful in doing so. We therefore constructed bacterial-like particles exhibiting the F and G proteins of NiV and observed their reactivity with two immunogenic antisera by immunoelectron microscopy and confirmed that the antisera were indeed reactive.

  1. English writing needs to be improved.

A: This manuscript has been revised by native English-speaking professionals.

Minor

  1. It is suggested to re-label Figure 1, Figure 2, and Figure 4 to improve the readability of the figure. so that the reader does not have to look for the notes repeatedly to understand the figures.

A: We have relabeled the corresponding figures.

  1. In Fig.3, “Anti-NiV F/G protein” should be marked as “Cy3”.

A: We have changed the “Anti-NiV F/G protein” to “Cy3” in Fig.3.

  1. Line 65, Line 241: IFA is usually an abbreviation of immunofluorescence assay. In this manuscript, authors adopted indirect immunofluorescence assay and direct immunofluorescence assay respectively. I suggest that iIFA and dIFA should be adopted here.

A: In lines 79, 89, 259, and 262: We have followed the suggestions and made the full text modifications.

  1. Line 175: “ELSIA” should be “ELISA”.

A: In line194: We have made the modification.

Reviewer 3 Report

Comments and Suggestions for Authors

The article "" has merit for publication in the prestigious journal "Vaccines". The authors have to incorporate the below suggested changes before the submission of final version of the article.

Line 18: Kindly change "are able" to "were able"

Line 22: Kindly change " candidate against" to " candidate vaccine against"

Line 57: Kindly change "plasmids that expressing" to "plasmids expressing"

Line 57: Kindly change "designated pSRV9-NiV-F" to "designated as pSRV9-NiV-F"

Line 95: Kindly change "2.1.5. Western blot" to "2.1.5. Western blotting"

Line 110: Kindly change "2.1.6. Transmission electron microscope" to "2.1.6. Transmission electron microscopy"

Line 116: kindly change "2.1.7. Immunoelectron microscope" to "2.1.7. Immunoelectron microscopy"

Line 123: Kindly change "antibody (pAB)" to "antibody (pAb)"

Line 379: Kindly change "the Th1-type cell immune. It has been shown that is usually" to "the Th1-type cell immune response. It has been shown that this response is usually"

Comments on the Quality of English Language

The English is OK.

Author Response

Line 18: Kindly change "are able" to "were able".

A: In line 18: We have made the modification.

Line 22: Kindly change " candidate against" to " candidate vaccine against"

A: In line 22: We have made the modification.

Line 57: Kindly change "plasmids that expressing" to "plasmids expressing"

A: In line 69: We have made the modification.

Line 57: Kindly change "designated pSRV9-NiV-F" to "designated as pSRV9-NiV-F"

A: In line 69-70: We have made the modification.

Line 95: Kindly change "2.1.5. Western blot" to "2.1.5. Western blotting"

A: In line 112: We have made the modification.

Line 110: Kindly change "2.1.6. Transmission electron microscope" to "2.1.6. Transmission electron microscopy"

A: In line 127-29: We have made the modification.

Line 116: kindly change "2.1.7. Immunoelectron microscope" to "2.1.7. Immunoelectron microscopy"

A: In line 133: We have made the modification.

Line 123: Kindly change "antibody (pAB)" to "antibody (pAb)"

A: In line 140: We have made the modification.

Line 379: Kindly change "the Th1-type cell immune. It has been shown that is usually" to "the Th1-type cell immune response. It has been shown that this response is usually"

A: In lines 396-97: We have made the modifications.

Reviewer 4 Report

Comments and Suggestions for Authors

Comments on the Quality of English Language

Please check and edit the language to improve the presentation of the manuscript.

Author Response

Major comments

  1. Title: Is it “systematic immune response” or “Systemic immune responses”? I guess the Systemic is a typo. “Systemic” is more appropriate since the authors have investigated cellular responses in the spleen and antibody responses in serum following immunization.

A: We have changed ' systematic immune response' to 'systemic immune responses'.

  1. Please interpret the results based on the statistical analysis if it is done for a particular data set. For example, in lines 303-306: Based on the statistics provided in Figure 9 graphs, TNF-α, IL-6, and IL-10 were significantly increased but not IFN-γ, IL-2, and IL-4 compared to the control group. Although all the cytokine levels were higher in treatment compared to control, if there is no statistical significance, it should not be considered biological significance. Also, in Figure 10A and line 308, IgG titer is significantly higher in the rSRV9-NiV-G group compared to the combination group at 56 days post-immunization based on the statistics indicated in the graph. It is not significantly higher compared to other groups in the rest of the time points. Therefore, please interpret the results based on the statistical significance, not by the visual observations. Please modify the sentences accordingly.

A: In lines 317-20 and 323: We have modified these sentences accordingly. And there is an error in Figure 9, we have made modifications.

  1. Line 228: Specifically describe what kind of test was used to analyze each type of data and what post hoc test was used in that situation OR indicate the statistical analysis (statistical method and post hoc test) performed for each data set in the figure legends to understand the results Eg. Figures 6, 7, 8, 9 and 10.

A: We have provided additional explanations in the figure legend.

  1. Discussion is not sufficient. Please discuss the results and not repeat the results in the discussion section. For example, the immunological reasoning behind observed results, association, or implication can be addressed. When describing the results, authors have made suggestions, inferences, or comments based on the results in the results section. These can be included in the discussion section, which helps constructively expand the discussion. The second paragraph of the discussion section is more appropriate for the introduction as it justifies the selection of vector and glycoprotein antigens of the Nipah virus in this study. Data produced by various assays shows the extent of input and the amount of effort in performing this study; however, the generated data is not discussed and highlighted enough to bring the significance of the results in the discussion section.

A: Based on your suggestions, we have expanded the discussion as much as possible.

  1. Please thoroughly check and edit the language.

A: This manuscript has been revised by native English-speaking professionals.

Minor comments

  1. Line19→It should be “responses”.

A: In line 19: We have made the modification.

  1. Line 20 & 389: Based on the result presented in Fig 11, it can be said that there is a reactivity between immunized sera and BLP displaying F and G antigens. Since it is not characterized quantitatively or qualitatively, the term “good reactivity” should be avoided.

A: In line 20: We have made the modification.

  1. Line 22: Please mention it as a vaccine candidate.

A: In line 22: We have made the modification.

  1. Line 47: I wondered about the reason for the selection of rabies vector in this study and later found it in the discussion section. As I mentioned above, it is most suitable to provide the justification in the introduction rather than in the discussion.

A: In lines 46-55: We have made the modifications as suggested.

  1. Line 58: Please spell “BSR cell” when using in the first time.

A: In line 70-71: We have made the modification.

  1. Line 61: It should be “The mixture of cells and supernatant were”.

A: In lines 74: We have made the modification.

  1. Line 68: Spell the abbreviations for the first time PBST and FITC

A: In lines 83-84: We have made the modifications as suggested.

  1. Line 92: Spell out “STE buffer”.

A: In line 108: We have made the modification.

  1. Line 108: Spell out “ECL color developer”.

A: In line 125: We have supplemented according to the suggestion.

  1. Line 144: Generally, 6-8 weeks-old mice are considered adults and used for experiments. Why do you use 4-6 weeks-old mice (young mice)? Why the Balb/c strain was selected over C57BL/6 mice?

A: We apologize that this is because our description was not detailed enough. In fact, the mice we purchased were 4-6 weeks old, but they were raised for 2 weeks to adapt to the environment before the experiment began. Thus, at the time of the official experiment, they were 6-8 weeks old. We have provided an additional description in Method.

Balb/c mice have a greater antibody response than C57BL/6 mice, making them better candidates for research on immune-related research. As a result, they are frequently utilized in studies to determine how immunogenic medicines and vaccinations.

  1. Line 155: What is the seeding density of cells in 96-well plates in the proliferation assay

A: In line 172: The density of cells in 96-well plates was 2.5×106 cells/mL.

  1. Line 164: What is the seeding density of cells, and how much volume of 0.25 μg/μL of rRABV was used in the assay?

A: In lines 182-83: Cell density was 2.5×106/mL, and a final concentrate of 0.25 μg/mL of F and G alone or in combination was added.

  1. Line 165: Change the term inoculation to incubation.

A: In line 189: We have made the modification.

  1. Line 168-169: 2.5*106/ml cells in 96 well ELISpot seems very high, and it is impossible to seed 96 wells with 1 ml of cells in this assay. Please correct it.

A: Rather of using 1 mL, we pipetted 200 μL of the 2.5×106 cells/mL splenocyte suspension into a 96-well plate.

  1. Line 172: Spell out “HRP”

A: In line 123-24: We have made the modification.

  1. Line 173: spell out “TMB” for the first time use.

A: In line 191: We have made the modification.

  1. Line 179: Spell out “BSA”.

A: In line 198: We have made the modification.

  1. Figure 2: Panel C is control. What type of control is this (unlabelled sample/non-specific binding control/ negative control etc)?

A: It is a negative control for virus uninfected cells and noted in the figure legend.

  1. Line 254-55 & Figure 5: If 18nm and 6nm gold particles were used to identify F and G, why are both antibodies visible in panel 5F, which is just rSRV9 without F or G?

A: In Figure 5F, RABV was recognized by the 18 nm gold-labeled antibody, and the 6 nm gold particle was non-specifically bound.

  1. Line 291: Replace “every 1 day” with “every day after....".

A: In line 305: The mice were weighed every other day after immunization. Therefore, we changed “every 1 day” to “every other day”.

  1. Line 303-304: What is the reason for selecting these cytokines? IL-1β and CXCL10 were reported to be correlated with protection in Nipah virus infection. Why were these not analyzed in this study?

A: In order to detect humoral immune parameters, the cytokines (IFN-γ, IL-2, and TNF-α) secreted by Th1-type cells and those (IL-4, IL-6, and IL-10) secreted by Th2-type cells were detected.

  1. Figure 6: I guess the virus titer in the Y-axis provided in the log scale. Please clearly indicate it in the graph or in the legend. IgTCID50 confuses with immunoglobulin short form Ig.

A: In line 299-300: We have added annotations in the figure legend.

  1. Figure 8B & 8C: IFN-γ & IL-4 spot numbers are significantly higher than control; however, this data does not correlate with Figure 9A & 9D (IFN-γ & IL-4) data as there is no statistical significance different from the control group. Any explanation for these observations? It seems there should be a statistical difference between groups in Figure 9D by visual observation. Please double-check the statistical analysis for this data.

A: The ELISApot and ELISA assay evaluated IFN-γ and IL-4 from both a cellular and humoral perspective. Therefore, the data results are different, but the trends are consistent. We have checked the statistical analysis and modified it accordingly.

  1. Figure 9B: rSRV9-NIV-F is higher with the smallest SD than rSRV9-NIV-G. However, rSRV9-NIV-G is indicated as statistically significant compared to the control, but no statistical difference is shown between rSRV9-NIV-F and the control. Please check the statistical analysis and correct it appropriately.

A: We have checked the statistical analysis and modified it accordingly.

  1. Figure 10A: I guess there should also be statistical significance at 35 days post- immunization since the control was almost closer to the X-axis. What type of statistical analysis was done with this particular data? Please check your statistical analysis.

A: Two-Way ANOVA was used for the statistical analysis of variance in the GraphPad Prism 8.0.1 software. The results of our statistical analysis were as follows.

  1. Is there any significant difference between the control and combination treatment group at 56 days post-infection since there is a difference between rSRV9-NIV-G and the combination group.

A: We double-checked the statistical analysis and there was no difference between rSRV9-NIV-G and the combination group.

  1. Figure 10B: Please indicate the time point in the legend for better understanding.

A: We have noted in the legend.

  1. Line 360: The Ebola virus, Lassa virus, and Zika virus are different viruses, not infectious diseases. Please modify the sentence for clear understanding.

A: We have made the modifications as suggested.

  1. Line 363: Change the word “higher mortality” to “high mortality”.

A: We have made the modification.

  1. Line 404: Will the data be available upon request?

A: Yes, we will provide data upon request.

Reviewer 5 Report

Comments and Suggestions for Authors

The paper entitled « Inactivated recombinant rabies virus displaying the Nipah virus envelope glycoproteins induces systematic immune response in mice” by Zhengrong Li et al presents the obtention of recombinant rabies viruses expressing either the Nipah virus envelope glycoproteins F or G, and their first characterization.

The experiments are well undertaken, properly described and the results are informative. Obviously it is a preliminary report towards a study which could demonstrate the efficiency of such vaccine for Nipah virus.

Several points should be considered:

·         Abstract: The authors are presenting data on inactivated rRABV. The choice of inactivation has to be discussed somewhere.

·         Introduction: Please discuss the advantage rRABV derived vaccine would have compared to the other systems

·         Figure 1 is not informative and the data could be presented in the methods

·         Figures 2 and 3 should be combined and please add a cartoon with a schematic representation of your rRABV sonstructs

·         Figure legends should be more detailed and the controls are not sufficiently described

·         Figures 6 and 4 and 5 should be combined. Figure 6 first. The Coomassie blue is not very informative with all the extra bands.

·         Figure 5 (D,E, F): Have you got better images showing the 6 nm gold particles?

·         Figures 8 and 9 and 10 should be combined.

·         Figure 7 could be omitted and the informations added in the text only.

·         Figure 11 and the text are difficult to understand. Please correct.

·         Discussion, line 352: “protective immunity”, the authors should considered to add such data to strengthen this paper has it has been undertaken by several groups already.

Author Response

  1. Introduction: Please discuss the advantage rRABV derived vaccine would have compared to the other systems.

A: We have added to the introduction accordingly.

  1. Figure 1 is not informative and the data could be presented in the methods.

A: We have deleted Figure 1.

  1. Figures 2 and 3 should be combined and please add a cartoon with a schematic representation of your rRABV constructs.

A: Figure 2 is direct immunofluorescence, while Figure 3 is indirect immunofluorescence. And, Figure 2 has not been stained for nuclei, so it is best not to combine the two figures.

In addition, we have added a schematic diagram of two constructs in which RABV expresses the F or G proteins of NiV and named it Figure 1.

  1. Figure legends should be more detailed and the controls are not sufficiently described.

A: We have described the figure legend and the control groups therein in as much detail as possible.

  1. Figures 6 and 4 and 5 should be combined. Figure 6 first. The Coomassie blue is not very informative with all the extra bands.

A: Although Figures 4-6 all identify recombinant RABVs, we believe it is preferable not to merge them, which would be logically clearer.

  1. Figure 5 (D, E, F): Have you got better images showing the 6 nm gold particles?

A: Despite our attempts to get clearer pictures, Figure 5D, E, F are already the best pictures under the electron microscope.

  1. Figures 8 and 9 and 10 should be combined.

A: Since these three figures used different methods to detect different immune indicators, the legend description would appear disorganized if they were combined. Therefore, we believe it is best not to combine them.

  1. Figure 7 could be omitted and the information added in the text only.

A: We think it is necessary to show the Figure 7.

  1. Figure 11 and the text are difficult to understand. Please correct.

A: We describe the legend for Figure 11 in more detail.

  1. Discussion, line 352: “protective immunity”, the authors should consider to add such data to strengthen this paper has it has been undertaken by several groups already.

A:In the Discussion section, we describe the reasons for failing to conduct protective immunization studies. This is indeed the greatest shortcoming of this study.

Round 2

Reviewer 2 Report

Comments and Suggestions for Authors

This revised version was much better than the first submitted version. I have no further comment.

Author Response

Thank you to the reviewer.

Reviewer 4 Report

Comments and Suggestions for Authors

Line 344: legend indicates the subclass of IgG in serum after 10 days of immunization. But, line 327 says it is after the last immunization (56 days post-immunization??). Please fix the discrepancy between these statements.

 Line 245: When we use ANOVA to test the data set, statistically significant results indicate that not all group means are equal. However, ANOVA results do not identify which groups differ from each other significantly. Post-hoc test or multiple comparison test is done following the ANOVA test to identify exactly which groups differ from each other. Therefore, please indicate what kind of post hoc test was used to analyze each type of data OR in  figure legends 6, 7, 8, 9, and 10.

Line 172 & 182: 2.5*106/ml cells were seeded in 96 well and 6 well plates. It doesn't seem very clear if someone wants to replicate the described method in this manuscript. Therefore, indicating the seeded cells as the "number of cells/well" is preferable.

Author Response

Line 344: legend indicates the subclass of IgG in serum after 10 days of immunization. But, line 327 says it is after the last immunization (56 days post-immunization??). Please fix the discrepancy between these statements.

A: We have made the modifications.

 Line 245: When we use ANOVA to test the data set, statistically significant results indicate that not all group means are equal. However, ANOVA results do not identify which groups differ from each other significantly. Post-hoc test or multiple comparison test is done following the ANOVA test to identify exactly which groups differ from each other. Therefore, please indicate what kind of post hoc test was used to analyze each type of data OR in figure legends 6, 7, 8, 9, and 10.

A: After ANOVA test, the Least Significance Difference (LSD) test was used.

Line 172 & 182: 2.5*106/ml cells were seeded in 96 well and 6 well plates. It doesn't seem very clear if someone wants to replicate the described method in this manuscript. Therefore, indicating the seeded cells as the "number of cells/well" is preferable.

A: In lines 172 & 182: We have made the modifications.

Reviewer 5 Report

Comments and Suggestions for Authors

Please take into account all of my previous comments. The article needs to be improved before publication.

Comments on the Quality of English Language

None

Author Response

  1. Introduction: Please discuss the advantage rRABV derived vaccine would have compared to the other systems.

A: We have added to the introduction accordingly.

  1. Figure 1 is not informative and the data could be presented in the methods.

A: We have deleted Figure 1.

  1. Figures 2 and 3 should be combined and please add a cartoon with a schematic representation of your rRABV constructs.

A: Figure 2 is direct immunofluorescence, while Figure 3 is indirect immunofluorescence. And, Figure 2 has not been stained for nuclei, so it is best not to combine the two figures.

In addition, we have added a schematic diagram of two constructs in which RABV expresses the F or G proteins of NiV and named it Figure 1.

  1. Figure legends should be more detailed and the controls are not sufficiently described.

A: We have described the figure legend and the control groups therein in as much detail as possible.

  1. Figures 6 and 4 and 5 should be combined. Figure 6 first. The Coomassie blue is not very informative with all the extra bands.

A: We have made the modifications.

  1. Figure 5 (D, E, F): Have you got better images showing the 6 nm gold particles?

A: Despite our attempts to get clearer pictures, Figure 5D, E, F (now Figure 4) are already the best pictures under the electron microscope.

  1. Figures 8 and 9 and 10 should be combined.

A: We have made the modifications.

  1. Figure 7 could be omitted and the information added in the text only.

A: We have deleted Figure 7.

  1. Figure 11 and the text are difficult to understand. Please correct.

A: We describe the legend for Figure 11 (now Figure 6) in more detail.

  1. Discussion, line 352: “protective immunity”, the authors should consider to add such data to strengthen this paper has it has been undertaken by several groups already.

A:In the Discussion section, we describe the reasons for failing to conduct protective immunization studies. This is indeed the greatest shortcoming of this study.

Round 3

Reviewer 5 Report

Comments and Suggestions for Authors

No further comments